# Initial Performance Analysis of the At-Altitude Radiance Ratio Method for Reflectance Conversion of Hyperspectral Remote Sensing Data

**DOI:** 10.3390/s23010320

**Published:** 2022-12-28

**Authors:** Luke J. R. DeCoffe, David N. Conran, Timothy D. Bauch, Micah G. Ross, Daniel S. Kaputa, Carl Salvaggio

**Affiliations:** 1Digital Imaging and Remote Sensing Laboratory, Chester F. Carlson Center for Imaging Science, College of Science, Rochester Institute of Technology, 54 Lomb Memorial Drive, Rochester, NY 14623, USA; 2Department of Electrical and Computer Engineering Technology, College of Engineering Technology, Rochester Institute of Technology, 15 Lomb Memorial Drive, Rochester, NY 14623, USA

**Keywords:** hyperspectral, sensor, calibration, radiance, reflectance, conversion, small unmanned aircraft systems, at-altitude radiance ratio, empirical line method, Headwall Nano-Hyperspec

## Abstract

In remote sensing, the conversion of at-sensor radiance to surface reflectance for each pixel in a scene is an essential component of many analysis tasks. The empirical line method (ELM) is the most used technique among remote sensing practitioners due to its reliability and production of accurate reflectance measurements. However, the at-altitude radiance ratio (AARR), a more recently proposed methodology, is attractive as it allows reflectance conversion to be carried out in real time throughout data collection, does not require calibrated samples of pre-measured reflectance to be placed in scene, and can account for changes in illumination conditions. The benefits of AARR can substantially reduce the level of effort required for collection setup and subsequent data analysis, and provide a means for large-scale automation of remote sensing data collection, even in atypical flight conditions. In this study, an onboard, downwelling irradiance spectrometer integrated onto a small unmanned aircraft system (sUAS) is utilized to characterize the performance of AARR-generated reflectance from hyperspectral radiance data under a variety of challenging illumination conditions. The observed error introduced by AARR is often on par with ELM and acceptable depending on the application requirements and natural variation in the reflectance of the targets of interest. Additionally, a number of radiometric and atmospheric corrections are proposed that could increase the accuracy of the method in future trials, warranting further research.

## 1. Introduction

Conversion of recorded radiance imagery to reflectance is essential to the production of accurate and useful information for a variety of remote sensing applications. The empirical line method (ELM) [1,2] is the most commonly used approach, which requires in-scene calibration panels with known spectral reflectance to be placed in the scene and imaged at various times during the collection event. This impacts the time and level of effort requirements of data collection, which are not shared with the alternative approach dubbed the at-altitude radiance ratio (AARR) [3,4,5,6].

While many previous studies have taken a modelling approach in characterizing the performance of AARR, this research focuses on real-world application; a miniature high resolution spectrometer was integrated into a newly designed sUAS-mountable downwelling irradiance sensor, coined the RIT Downwelling Irradiance Sensor (RIT DIS), in order to collect the data required for AARR. The initial version of the RIT DIS has produced encouraging results for AARR with multispectral sensors, where the MicaSense RedEdge-M multispectral camera was used to capture scene radiance [7]. This paper aims to extend these findings to hyperspectral data by instead using the Headwall Nano-Hyperspec to record scene radiance. The Nano-Hyperspec and the RIT DIS are mounted on a DJI Wind8 sUAS and a target scene containing several reflectance panels is imaged at a low altitude of ~30 m, and a higher altitude of ~90 m. Additionally, ELM provides a baseline for reflectance conversion performance comparison.

While ELM was previously shown to slightly outperform AARR [7], the data were collected under a clear sky, which is one of the known best case scenarios for the ELM method. ELM cannot account for changes in illumination conditions unless the in-scene calibration panels are imaged regularly throughout the flight, which introduces logistical issues for many data collection events and thus it is not commonly feasible. These changing conditions between ELM calibrations can result in a substantial impact to the recorded scene radiance, confusing subsequent analysis results. As the AARR method captures downwelling irradiance data in real time throughout the collection event, it has the ability to adapt to these changing conditions. Accordingly, the data for this study were collected across several days with increasingly challenging weather conditions in order to determine the use cases where AARR would be preferred over ELM. Furthermore, the importance of this study is further highlighted by considering the ease of data collection and the automation potential granted by AARR. Real-world applications of AARR has been researched previously, including downwelling corrections for platform attitude and atmospheric effects, but like other studies, data were collected on a clear day [8,9]. This study focuses on a more rudimentary approach to AARR in order to deduce findings that can support its use under a variety of weather conditions.

This paper is organized into the following sections; Section 2 discusses the materials and methods used in further detail, including reflectance conversion and methods in Section 2.1, Section 2.2 and Section 2.3, the specifications and particulars of the sensors used in Section 2.4, the data collection process in Section 2.5, the post processing pipeline in Section 2.6 and the tools used for the analysis of the recorded data in Section 2.7. Finally, qualitative results are presented in Section 3 and discussed in Section 4.

## 2. Materials and Methods

### 2.1. Reflectance Conversion Preliminaries

Remote sensing data are captured in raw digital count by the sensor and is then commonly converted to radiance using a variety of radiometric calibration techniques. Reflectance conversion is the next step in the processing chain. It allows practitioners and scientists in a variety of fields to discern further important details about specific materials in the target scene. As mentioned previously, the most popular method to perform this conversion is the empirical line method (ELM), however the at-altitude radiance ratio (AARR) promises to facilitate the data collection process by reducing the number of personnel and on-site equipment required. Therefore, it has the potential to progress the automation of the data collection process for sUAS-based remote sensing.

Before exploring these reflectance conversion methods in the following sections, another important element to consider is the bidirectional reflectance distribution function (BRDF) and the impact it has on reflectance measurements. BRDF represents the reflectance factor of a particular material as a function of illumination angle, detector imaging angle and other environmental factors [10]. A portable goniometer was created to measure the BRDF of a material in the field to facilitate extensive research of BRDF for a multitude of imaging applications [11], which emphasizes the importance of BRDF consideration.

Since the reflectance factor of materials is a function of illumination angle and detector imaging angle, it must be considered in all reflectance measurements. In sUAS-based remote sensing, the imaging angle is most often at nadir (zenith relative to the material). It should be noted that the field of view (FOV) of the sensor on the platform will have a direct impact on the imaging angle and will cause it to vary for each recorded pixel. The attitude of the platform is also changing throughout the flight as it tries to stabilize, thus the imaging angle is not perfectly nadir across the sensor’s aperture. Yet, the illumination angle of the target surface varies as the solar zenith angle changes throughout the collect. Consequently, it is commonplace for remote sensing reflectance conversion methods to make the assumption that the diffuse reflectance is the main component being measured, ignoring the specular component. This assumption implies that the materials in the target scene have a constrained BRDF, independent of detector imaging angle, which is a property of Lambertian surfaces. Although nearly all materials on the Earth’s surface are not Lambertian, the actual BRDF and an assumed Lambertian BRDF of materials are consistently in agreement when the solar zenith angle is between 30 and 45 degrees and the imaging angle is at zenith [10,12,13]. The error introduced by this assumption is inherent to both ELM and AARR, furnishing a fair ground for comparison between the two.

It is important to note that radiance reflected by the specular component of reflectance must come from a particular direction and might not always be from direct sunlight; they can come from specular reflections from a variety of objects such as leaves blowing in the wind, windows, sand grains and wave facets in the water. Specular reflections from direct sunlight will be much brighter than specular reflections from surrounding objects, but based on the nearly nadir imaging angles typical of sUAS-based sensors and the position of the sun in the sky during these collects, true specular reflections from direct sunlight should not contribute to measured radiance. The specular reflections from other objects in or near the scene can and will affect radiance measured by the sensor, but they should only affect the recorded radiance of certain pixels in the scene, not all, as they are typically less than fraction of a pixel in size. On average, the recorded radiance of a target within the scene across all pixels will be less likely to be affected by these specular reflections, which further explains why the omission of the specular component of reflectance is reasonable.

### 2.2. Empirical Line Method

ELM identifies a linear relationship between radiance and reflectance and uses it as a basis for reflectance conversion [1,2]. This linear relationship holds true between digital count and reflectance as well assuming the detector response is linear, allowing the use of ELM without calibrating the sensor for radiance. Although this may be beneficial to some, remote sensing practitioners typically prefer a radiance calibrated sensor where detector non-linearity is corrected for [14].

ELM requires a number of calibration panels to be placed in the target scene, and a field spectrometer is used to measure the reflectance of these panels prior to the data collection event. Before measuring the in-scene panels, the field spectrometer is calibrated for reflectance by first measuring the radiance of a white reference panel. This panel must be clean, in pristine condition and have calibrated reflectance measurements supplied by a provider such as Labsphere, Inc. (North Sutton, NH, USA) for a Spectralon^®^ panel. The reflectance measurements of the in-scene calibration panels can then be recorded. It is important to note that other reflectance measurements of in-scene calibration panels can be used as a basis for this second approach, such as those measured in a lab setting or on previous collect days; however, it is common practice to take measurements in the field on the same day since they can be substantially influenced by changes in illumination conditions, as discussed in Section 2.1.

With the reflectance measurements of the in-scene calibration panels in hand, the collection flight can commence. The sUAS-platform must image these panels at least once or at various points during the flight which allows their digital counts or radiance values to be extracted during post processing. Any realistic number of panels of varying spectral reflectance can be used to determine the ELM relationship; commonly one or two panels are used, referred to as 1-point ELM and 2-point ELM, respectively. For 3 or more calibration panels, a line of best fit is applied using an assortment of regression techniques. For this research, 1-point and 2-point ELM are used as references for our experiments as they are currently the preferred methods for reflectance conversion; arbitrary examples are shown in Figure 1.

For 2-point ELM, two panels are deployed in the field, one dark and the other bright. The relationship for 2-point ELM can then be calculated using Equations (Equation 1)–(Equation 3) [1,2].
(1)$$\rho_{\lambda }\left(x,y\right)=m_{\lambda }L_{s,\lambda }\left(x,y\right)+b_{\lambda }\;,$$
(2)$$m_{\lambda }=\frac{\rho_{\mathrm{bright},\lambda }-\rho_{\mathrm{dark},\lambda }}{L_{s,\mathrm{bright},\lambda }-L_{s,\mathrm{dark},\lambda }}\;,$$
(3)$$b_{\lambda }=\rho_{\mathrm{bright},\lambda }-m_{\lambda }L_{s,\mathrm{bright},\lambda }\;,$$
where $\rho_{\mathrm{bright},\lambda }$ is the average reflectance of the bright target, $\rho_{\mathrm{dark},\lambda }$ is the average reflectance of the dark target, $L_{s,\mathrm{bright},\lambda }$ is the average radiance or digital count of the bright target, $L_{s,\mathrm{dark},\lambda }$ is the average radiance or digital count of the bright target, *x* and *y* denote the pixel coordinates and λ denotes the center wavelength of the spectral band. It should be noted that the bias, $b_{\lambda }$, can be computed using either the dark or bright target, and represents the path radiance, or backscattering, between the target scene to the imaging platform. Essentially, this is the solar scattered spectral path radiance generated in the path between target and the sensor [4].

For 1-point ELM, only one bright panel is placed in the scene. Since two points are required to compute a linear relationship, $b_{\lambda }$ from Equation (Equation 3) is assumed to be zero, which requires the assumption that $L_{s,\lambda }\left(x,y\right)=0$ implies $\rho_{\lambda }\left(x,y\right)=0$. The 1-point ELM relationship then simplifies to Equations (Equation 4) and (Equation 5) [1,2].
(4)$$\rho_{\lambda }\left(x,y\right)=m_{\lambda }L_{s,\lambda }\left(x,y\right)\;,$$
(5)$$m_{\lambda }=\frac{\rho_{\mathrm{bright},\lambda }}{L_{s,\mathrm{bright},\lambda }}\;,$$
where $\rho_{\lambda }\left(x,y\right)$ is the ELM calculated reflectance, $L_{s,\lambda }\left(x,y\right)$ is the at-sensor radiance recorded from the scene, $L_{s,\mathrm{bright},\lambda }$ is the spatially averaged radiance of the bright panel extracted from the radiance imagery recorded by the sensor, $\rho_{\mathrm{bright},\lambda }$ is the measured reflectance of the bright panel, *x* and *y* denote the pixel coordinates and λ denotes the center wavelength of the spectral band.

For 1-point ELM, the assumption of zero path radiance implies that $L_{s,\lambda }\left(x,y\right)=0$ when observing a target with $\rho_{\lambda }\left(x,y\right)=0$. Figure 2 shows 1-point and 2-point ELM relationships calculated with real data under clear conditions at an altitude of 90 m for a chosen wavelength of 555 nm. For 2-point ELM, the path radiance was non-zero and estimated to be 0.0008 W/m^2^-nm-sr. This causes 1-point ELM to overestimate the reflectance of weak reflectors by a maximum of 0.002 reflectance units and underestimate the reflectance of strong reflectors by a maximum of 0.004 reflectance units. Thus, the assumption should not have a substantial impact on ELM-estimated reflectance under clear conditions at typical sUAS altitudes.

The widespread use of ELM in remote sensing research is easily explained by its reliable performance and relatively low error [1,2]. However, it has many drawbacks, including the setup time needed to prepare on-site equipment, the requirement to re-image the calibration panels at multiple times during the collect and the decrease in accuracy during changing illumination conditions. These issues are undesirable as they ultimately reduce the time effectiveness and defer the furtherance in automation of the data collection process, which could be avoided by the development and research of the AARR approach.

### 2.3. At-Altitude Radiance Ratio (AARR)

AARR, a more recently proposed method for reflectance conversion [3], is the focus of this research. The derivation of the simplified AARR equation begins with the effective at-sensor radiance $L_{s,\lambda }$ for the reflective/VIS-NIR region (400–1000 nm) at Equation (Equation 6), in which all energy paths of thermal emission are negligible [15].
(6)$$L_{s,\lambda }=\left[E_{\mathrm{TOA},\lambda }\tau_{↓,\lambda }\cos\sigma +E_{\mathrm{sky},\lambda }\right]\frac{\tau_{↑,\lambda }\rho_{d,\lambda }}{\pi }+L_{a,\lambda }$$
where $E_{\mathrm{TOA},\lambda }$ is the exo-atmospheric solar irradiance, $\tau_{↓,\lambda }$ is the transmission through the atmosphere to the target, σ is the solar zenith angle, $E_{\mathrm{sky},\lambda }$ is the hemispherical diffuse irradiance at the Earth’s surface, $\tau_{↑,\lambda }$ is the path transmission from target to sensor, $\rho_{d,\lambda }$ is the target’s Lambertian reflectance and $L_{a,\lambda }$ is a combination of path radiance and adjacency effects.

Since sUAS-based remote sensing typically operates at low altitudes between 30 and 90 m, the path radiance will contribute negligibly to the total at-sensor radiance [3], as long as the wavelength dependent optical depth between the ground and the sensor is close enough to 0, which would not be the case in foggy or dusty conditions. Additionally, adjacency effects are negligible for scenes that are mostly horizontal without tall objects in the vicinity [16]. Reconsidering the previous assumption that the imaging platform is operating below altitudes of 90 m, it follows that $\tau_{↑,\lambda }$ can be expected to be very close to unity. The AARR relationship, Equation (Equation 7), is formulated by applying these assumptions to Equation (Equation 6) and rearranging to solve for $\rho_{d,\lambda }$
(7)$$\rho_{d,\lambda }=\pi \frac{L_{s,\lambda }}{E_{\mathrm{TOA},\lambda }\tau_{↓,\lambda }\cos\sigma +E_{\mathrm{sky},\lambda }}$$
where the denominator of Equation (Equation 7) is simply an approximation of the downwelling irradiance at the Earth’s surface, which can be measured and exploited for reflectance calculations by the implementation of a downwelling irradiance spectrometer [3]. Since this sensor needs to capture hemispherical downwelling irradiance, a cosine corrector with a 180° FOV is attached. This measurement, denoted as $E_{\mathrm{DIS},\lambda }$, simplifies Equation (Equation 7) to Equation (Equation 8).
(8)$$\rho_{d,\lambda }=\pi \frac{L_{s,\lambda }}{E_{\mathrm{DIS},\lambda }}$$

In practice, AARR can be applied using data collected by a ground-based sensor [7]. However, this does not alleviate the inconveniences of the ELM method noted in Section 2.2, and furthermore a fixed-in-place, ground-based sensor cannot realistically be in the area being imaged at all times. Instead, the downwelling irradiance sensor can be mounted on the imaging platform, placing it directly above the target scene, so any changes in illumination conditions are captured in real time. In-scene reflectance panels would not be required, and would not have to be imaged throughout the flight, substantially reducing the level of effort required for initial data collection setup and subsequent data analysis.

To make this work, the $E_{\mathrm{DIS},\lambda }$ term in Equation (Equation 8) is assumed to be equivalent to the surface irradiance in the target scene. Although, this is not always the case; cloudy conditions may cast shadows over specific areas of the scene but not others, or on the downwelling irradiance sensor itself. Figure 3 illustrates these situations, but currently the AARR method assumes that irradiance variations on the ground are the same as at the sUAS. By extension, this assumption disregards the absorption and scattering in the two-way imaging path between the platform and the surface, building on the assumptions made in the derivation of the AARR equation. Thus, it is expected that measurement errors will increase with altitude. The increase of this error between 15 and 100 m of altitude was shown to be less than 0.01 absolute reflectance units for multispectral remote sensing [7], which is negligible when considering the reflectance variability of vegetation [5]. Nonetheless, the data used in this paper were collected at low and high altitudes to verify these findings for hyperspectral imaging.

### 2.4. Sensors

The Nano-Hyperspec and RIT DIS used in this research to collect at-sensor radiance and downwelling irradiance, respectively, were introduced in Section 1. They are integrated onto the MX2 sensor suite mounted on a DJI wind8 sUAS, which allows each sensor to capture data simultaneously. The MX2 sensor suite is a multimodal payload built at RIT, and includes Multispectral, Hyperspectral, RGB, SWIR, LWIR and Lidar sensors that can record data simultaneously. In order to properly analyze the performance afforded by AARR with this arrangement and support the use of 2-point ELM as a direct comparison, the Spectra Vista Corporation (SVC) HR-1024i Field Spectrometer is used to measure surface reflectance of various in-scene targets. This section outlines the specifications and calibration details of each of these three sensors.

#### 2.4.1. Headwall Nano-Hyperspec

The Headwall Nano-Hyperspec, a push-broom hyperspectral sensor, was used to capture the scene radiance data for this research. The Nano is calibrated for radiance at Headwall. The camera is depicted in Figure 4 and its specifications are listed in Table 1. The F/2.5 fore-optic has a 12 mm focal length coupled with the 7.4 μm pixel pitch results in an imaged FOV of 22.6°, meaning the imaging angle for pixels at the periphery of the detector array would be as much as 11.3° assuming the sensor is pointing perfectly nadir. Before a flight, the integration time of the sensor is set using a white reference panel. This panel is made of Spectralon^®^, a cintered polytetrafluoroethylene (PTFE) developed by Labsphere, Inc. in 1986. It has an almost uniform spectral reflectance of ~50% from 400–1500 nm and exhibits Lambertian properties [17]. A target with this particular reflectance is used since most of the targets of interest exhibit ~50% reflectance, so optimizing the integration time for these targets can help to mitigate saturation.

#### 2.4.2. RIT Downwelling Irradiance Sensor

The RIT DIS, a new sUAS-mounted sensor system, was built to measure and record downwelling irradiance throughout a collection flight. It enables the application of the AARR method with any radiance-calibrated sensor that is responsive in the VIS-NIR range. It was first applied to multispectral imagery [7], and is pictured in Figure 5. The RIT DIS records spectral irradiance measurements once every second synced with an integrated GPS, allowing each capture to be accurately timestamped. The integration time of the spectrometer is optimized prior to recording any data, and is then optimized at the frequency and saturation level set by the user.

The RIT DIS has been lightly modified since its use in the previous multispectral study. The specifications of the main component, the Ocean Insight (formerly Ocean Optics) Flame-S-VIS-NIR-ES spectrometer, are listed in Table 2. An Ocean Insight CC-3-DA direct-attach cosine corrector is attached to the aperture of the spectrometer, providing a theoretical 180° FOV. However, the shape of the fore-optic may limit the actual FOV to approximately 168°, as calculated by measuring its geometry. The spectrometer was radiance calibrated using a Labsphere, Inc. 20-inch integrating sphere, which is not how these instruments are typically calibrated. A desirable advantage of this calibration method is that the sensor’s foreoptic does not need to be characterized as long as it remains unchanged when used for field measurements. Further, this method does not account for cosine fall off as the calibration is carried out using Lambertian illumination, which is useful if the illumination during data collection is uniform, such as on an overcast day. However, this type calibration can induce errors when directional point sources of light are present, such as the sun on a clear day. Additionally, since the cosine corrector is assumed to be Lambertian, radiance measurements can be converted to irradiance, and vice-versa, by applying a factor of π.

The RIT DIS is mounted to the top of the sUAS at a fixed angle of 4° aft of zenith, which offsets the common forward pitch of the platform during forward flight while collecting data, keeping the center of the cosine corrector’s FOV as close as to zenith as possible. This angle was chosen carefully by analyzing historical attitude data from this specific sUAS. It is important to note that the attitude of the platform can vary significantly during the collect since its stability is affected by a variety of flight conditions. Since the sensor is mounted at a fixed position, the sensor cannot account for these changes in platform attitude. The effects of this design on downwelling irradiance measurements were explored and are documented in Appendix A.

#### 2.4.3. SVC HR-1024i Field Spectrometer

The HR-1024i, depicted in Figure 6, is a commonly used portable field spectrometer; its specifications are listed in Table 3. The spectral reflectance of various in-scene targets was measured by the HR-1024i before each data collection flight, in order to discern a baseline reflectance measurements for both the application of 2-point ELM and the comparison with each conversion method. Since this sensor takes a white reference panel measurement before measuring each target, it does not need to be calibrated for radiance to sample reflectance. However, the precision of these reflectance measurements must be quantified in order to adequately compare them with the reflectance estimates produced by ELM and AARR. Appendix B summarizes the precision analysis of the HR-1024i.

### 2.5. Data Collection Process

As expressed previously, data were collected over three days with varying illumination conditions in order to assess the robustness of each reflectance conversion method. During each day, two flights were completed; one at 30 m and another at 90 m of altitude. All data collection events were completed at the Tait Preserve in Penfield, NY (43°8${}^{\prime }$27.6${}^{\prime \prime }$ N, 77°30${}^{\prime }$21.6${}^{\prime \prime }$ W). A total of seven target panels were placed in the scene, approximately 3 m apart, to act as regions of interest (ROIs). The panels are made of plywood covered in colored felt, with one red, one blue, one green, and four shades of gray. The spectral reflectance of each of these ROIs was measured by the SVC HR-1024i at the beginning of each collect day. Figure 7 displays the arrangement of this scene.

### 2.6. Post Processing Pipeline

For this study, reflectance imagery needed to be generated using AARR and both 1-point and 2-point ELM. For AARR, the first step in the post collection processing pipeline is to build up a hypercube of downwelling irradiance measured by the RIT DIS for each radiance hypercube recorded by the Nano-Hyperspec. As seen in Section 2.4, the two instruments have different spectral channels and pixel dispersions, meaning the higher resolution spectrum of the downwelling spectrometer needs to first be interpolated to the spectral channels of the Nano-Hyperspec. However, without accounting for the differences in detector response characteristics before interpolation, artifacts can be created by acute spectral features present in the higher resolution data. A rudimentary analysis was required to transfer the spectral resolution of one instrument into the other.

Spectral emission lamps with samples of Argon (Ar) and Mercury-Neon (HgNe) were used to derive a Gaussian kernel approximation to the blurring inherent to the lower resolution of the Nano-Hyperspec. The spectral response curves of isolated spectral lines in the samples were averaged to determine a simple blur kernel that can be convolved with the downwelling spectrum, in an effort to diminish the effects of the acute spectral features caused by the spectral resolution differences between the two instruments. This kernel is a simple Gaussian fitted to the average spectral response curve, resulting in a full width at half maximum (FWHM) of ~4.5 nm; the effects of this blur are illustrated in Figure 8.

This analysis also identified significant errors in the wavelength calibration of the Nano-Hyperspec. To correct this error so that both instruments were in agreement, the same spectral emission lamps were also measured by the RIT DIS. These allowed a wavelength shift parameter to be determined from the wavelength errors of each instrument, which helped increase the accuracy of wavelength matching between them.

In practice, this wavelength shift parameter was applied to the upwelling channels, the blur kernel was convolved with the downwelling irradiance spectrum and then the interpolation of the upwelling radiance spectrum to the shifted channels followed. As mentioned, this is a rudimentary approach to this problem, but the derivation of a transfer function between the spectral response of each of the instruments would be the ultimate solution. This has been previously researched for reflectance measurements using the AARR approach when operating instruments with comparable spectral profiles [18], but needs to be explored for contrasting sensors as further discussed in Section 4.

Following interpolation, each line of the upwelling radiance hypercube is matched with the analogous downwelling irradiance spectrum using the timestamps provided by the internal GPS of each instrument, achieving a downwelling irradiance hypercube for each upwelling image. It should be noted that while the Headwall software does produce ortho-rectified imagery, this processing must be done using the raw data; the transfer function for ortho-rectification is not provided to the end-user, limiting the ability for each pixel to be accurately mapped to the downwelling spectrum that was recorded at the same time. AARR can then finally be performed using Equation (Equation 8), which is essentially a simple division between the upwelling and downwelling hypercubes, resulting in a hypercube image of the target scene in reflectance units.

For ELM, reflectance imagery is generated by Headwall’s software using 1-point ELM, so additional processing steps were only required for 2-point ELM. This task is initiated by extracting calibration panel data from one of the first Nano-Hyperspec radiance images. The lightest gray panel and the black panel were used as bright and dark targets, respectively. The radiance data for each panel are then averaged spatially, and the corresponding reflectance data measured by the SVC HR-1024i are interpolated to the wavelengths of the Nano-Hyperspec. This does not require a transfer function like the RIT DIS does, since the reflectance spectrum is smooth and slowly varying. Together, these are used to form the 2-point ELM relationship in Equations (Equation 1)–(Equation 3), which is formatted into a function that accepts upwelling radiance hypercubes. Each radiance hypercube recorded by the Nano-Hyperspec during the collect can then be entered into the ELM function, generating reflectance products.

### 2.7. Data Analysis

Prior to statistical analysis, the measured reflectance of each of the in-scene target panels must be extracted reflectance imagery generated by the methodology described in Section 2.6. A hyperspectral image annotation tool that allows the data from each reflectance conversion method to be extracted simultaneously was developed as part of this research, in an effort to reduce the level of effort required for this task. Once extracted, the distribution of the data for each panel across each conversion method is then compared to the reflectance measured on the ground by the SVC HR-1024i prior to the flight using a generalization of the $t^{2}$ statistic from the Student’s t-distribution to the multivariate case [19], beginning with Equation (Equation 9). This tests the null hypothesis $H_{0}:\mu =\mu_{0}$, where μ represents the actual mean of the population being sampled, which is the mean reflectance of the target generated by the conversion method, and $\mu_{0}$ is the panel spectral reflectance measured on the ground by the SVC HR-1024i.
(9)$$T^{2}=n{\left(\bar{X}-\mu_{0}\right)}^{T}S^{-1}\left(\bar{X}-\mu_{0}\right)$$
where $T^{2}$ is the multivariate generalization of $t^{2}$ statistic, $\bar{X}$ is the mean spectral reflectance of the panel extracted from the reflectance imagery generated by the conversion method under comparison and $S^{-1}$ is the covariance matrix of that data. $T^{2}$ is an appropriate performance indicator for this research as it measures how far the mean of the sampled spectral reflectance is from the reflectance measured by the field spectrometer, while taking into account the spectral correlations and variances among the samples. Equation (Equation 10) shows the proportional relationship between $T^{2}$ and an *F*-distribution with *p* and n−p degrees of freedom, where *p* is the length of the reflectance vector and *n* is the number of samples.
(10)$$T^{2}∼\frac{(n-1)p}{\left(n-p\right)}F_{p,n-p}$$ This test will reject the null hypothesis, $H_{0}$, when Equation (Equation 11) is satisfied,
(11)$$T^{2}\geq c_{0}=\frac{(n-1)p}{\left(n-p\right)}F_{p,n-p}\left(\alpha \right)$$
where α is a number between 0 and 1, corresponding to the upper 100α percentile of the *F*-distribution. For the purposes of this study, a common selection of α=0.95 was chosen. This choice effectively makes the associated *p*-value drop below 0.05 when $T^{2}\geq c_{0}$, which indicates a statistically significant difference, hence the null hypothesis $H_{0}$ would be rejected.

Additionally, since the $T^{2}$ statistic is heavily weighted on the variance of the samples, the average root mean square error (RMSE) in reflectance units is calculated as well using Equation (Equation 12).
(12)$$\mathrm{RMSE}=\sqrt{\frac{\sum_{i=1}^{n}{\left({\bar{X}}_{i}-\mu_{0_{i}}\right)}^{2}}{n}}$$

## 3. Results

In this section, the results are separated by each of the three data collection days, labelled by their corresponding sky conditions. Each section describes the conditions during the collect in further detail, indicating important data such as the data and time, solar zenith angle, and cloud types that were present in the sky. There was no precipitation present during any of the collects. The figures show the distributions of the sampled reflectance using each conversion method with separate figures for each in-scene panel, and indicate those distributions using a 2-standard deviation envelope. The three tables in each section present both the $T^{2}$ statistics and RMSEs; one covers all collected data for the day while the other two cover data subdivided by collection altitudes of 30 and 90 m relative to the ground. Red values indicate if the $T^{2}$ statistic rejects the null hypothesis. The collected data are analyzed in the region from 400 to 900 nm on account of substantial uncorrected non-linearity effects present in the RIT DIS downwelling irradiance data in the region above 900 nm, which will be addressed in future work. It is important to note that the ELM reflectance measurements are based on a relationship derived using the first radiance image of the scene in order to capture the variance caused by changes in illumination conditions during the collect.

As mentioned previously in Section 2.1 and Section 2.4.1, the FOV and attitude of the sensor play into the effective imaging angle for each recorded pixel. Figure 9 is a heat map of the effective imaging angle for pixels containing samples of target panels across all three data collects, where the average imaging angle was 1.41°. As expected, most samples were recorded at non-nadir imaging angles, however the majority of the samples were recorded at imaging angles within 5° of nadir.

### 3.1. Light Clouds

The environmental conditions for data collection were not ideal; the non-uniform dispersion of light cloud layers in the atmosphere are typically undesirable. These conditions are presented visually in Figure 10 and qualitatively in Table 4. Table 5 and Figure 11 and Figure 12 present the overall results for each method during this collect, which show that the null hypothesis is not rejected for all methods across all panels. RMSEs for AARR range from ±0.017 to ±0.051, RMSEs for 1-point ELM range from ±0.020 to ±0.064 and RMSEs for 2-point ELM range from ±0.016 and ±0.055 amongst the seven targets.

Table 6 and Table 7 break down the results at altitudes of 30 and 90 m, respectively, which again shows that all conversion methods passed the $T^{2}$ hypothesis test. The RMSEs observed by all three methods seemed to decrease with altitude for all targets except the medium gray panel, which actually increased for all three methods. The highest RMSEs were observed on the red and light gray panels, which have the highest reflectance amongst the group. The RMSEs for black panel, the panel with the lowest overall spectral reflectance, increased by approximately ten-fold across each method at the higher altitude.

### 3.2. Scattered Clouds

The data collected on this day represented atypical conditions for remote sensing as a variety of cloud layers were present, as depicted in Figure 13 and Table 8. These cloud layers were dispersed in inconsistent concentrations, which resulted in complex and ever-changing illumination levels. It must be noted that ground-reference reflectance measurements were taken under clear conditions beforehand, hence may impact the results presented in this section. Qualitative results are shown in Figure 14 and Figure 15, while quantitative results shown in Table 9, Table 10 and Table 11 show multiple cases where the null hypothesis is rejected; however, these cases are not always correlated with higher RMSEs.

Interestingly, all three methods seemed to struggle in measuring the reflectance of the red, blue and light gray panels, while they all observed an average RMSE of ±0.022 amongst the other four.

### 3.3. Overcast

Conditions during the final data collection for this paper presented mainly diffuse illumination on account of the uniformity of the overcast sky, presented in Figure 16 and Table 12. The plots in Figure 17 and Figure 18 show large agreement between predicted and ground-measured reflectance, yielding low variance and relatively high accuracy. The majority of the results reported on this day feature low RMSEs, despite some cases observing a rejection of the null hypothesis at higher altitude, seen by cross-referencing Table 13, Table 14 and Table 15.

## 4. Discussion

While the results of the $T^{2}$ tests and calculated RMSEs can be used as a relative indicator of the overall performance of AARR and ELM in each use case, it must be surmised that the absolute accuracy and precision of these methodologies is difficult to isolate and compare due to variations in atmospheric conditions and their affects on reflectance measurements used as ground references for statistical anaylsis. As discussed in Appendix B, this is inherent to all field measurements, but more can be done to provide better ground-reference reflectance data for calculation of these measurement errors. The variation of the BRDF of the felt-covered panels used in this study most likely contributes to the errors observed by reflectance conversion techniques, so a progression to the use of a variety of shades of Permaflect^®^ or other panels with consistent BRDFs is suggested for following studies.

Examining the $T^{2}$ and RMSE values observed for each method, the performance of all methods show a decrease at higher altitude on average, however this decrease with altitude is more apparent with AARR, understandably by its definition. The decreased performance with altitude is less apparent on the overcast day, especially for ELM, which stays reasonably consistent. In fact, the results observed during overcast conditions for both AARR and ELM confute the common reservations among remote sensing practitioners that have lead to the avoidance of data collection under these conditions, which has also been brought up in previous studies [5]. Certainly, lower illumination levels can lead to lower signal-to-noise ratios (SNR) within a detector, however the results clearly show high precision in reflectance measurements and acceptable levels of error for many remote sensing applications, as the highest RMSE observed was only ±0.033. For AARR, the lower error on the overcast day can be attributed to the calibration methodology, as the uniform illumination conditions present matched those during calibration using the integrating sphere. Contrastingly, this resulted in higher errors during days with the sun exposed, resulting in a point source dependence as expected.

Further, the observed RMSEs for AARR were on par with ELM, and are oftentimes lower depending on the panel in question. For most cases, except for a select few for the red and light gray panels, the RMSE of AARR reflectance measurements are below ±0.10, with a large majority below ±0.050 and many below ±0.020. This is well within the reflectance variation of many materials commonly analyzed in remote sensing such as vegetation [5]. Although AARR performed similar to ELM overall, variations and inaccuracies in the recorded downwelling irradiance can cascade into the reflectance produced by AARR. This is almost analogous to how variations in illumination conditions affect reflectance estimated by ELM. However, unlike ELM there are a variety of corrections that can lead to the reduction of the variance observed in AARR, which will improve its precision.

As discussed in Appendix A, the zenith angle of the sensor can have a large impact on the downwelling irradiance measurement, especially under non-uniform conditions. This can be addressed by angular correction, which can be done in multiple ways. The least feasible option, due to weight restrictions and effect on flight characteristics of an sUAS, is to use a gimbal in an effort to keep the sensor at zenith at all times. Another approach is to first characterize the response of the cosine corrector as a function of zenith and azimuth angles, which would also help to account for the solar zenith and azimuth angles, and then use that response to derive a correction based on the attitude of the platform [20]. Further, a cosine corrector with a more uniform response across azimuth should be used or designed, which is particularly important when the sun is unconcealed and when non-uniform illumination is present. There are other approaches that have been researched previously to correct for off-zenith angles [21,22], however they use additional sensors which can add to the complexity and weight of the overall system.

The radiometric calibration of the spectrometer used in the RIT DIS can be improved, as the effects of non-linearity, stray light, and exposure time have been left largely uncorrected. Non-linearity, dark current and exposure time effects could be corrected through photon transfer curve characterization, and will have a significant impact on the accuracy of the detector. Changes in detector behaviour with temperature should be analyzed and corrected for, as it will also have underlying impacts on the radiometric calibration. Further, atmospheric corrections can be developed both physics-based approaches [13] and those based on modelling and simulation of AARR [3,6,23] can lead to lower observed errors.

Additionally, there are areas outside of downwelling measurement that could help improve the performance of AARR. The Nano-Hyperspec used in this study will be better calibrated for radiance, as there is a tremendous amount of ongoing work to characterize every aspect of the sensor, including flat-fielding and smile correction. As discussed in Section 2.6, the development of a transfer function between hyperspectral sensors with dissimilar response functions and an ortho-rectification technique for matching AARR downwelling irradiance to upwelling radiance are both vital to the progression of AARR.

In summary, the observed error introduced by the AARR method was shown to be on par with ELM and acceptable depending on the application requirements and natural variation in the reflectance of the targets of interest. The vastness of areas for improvement for the AARR technique is compelling for future research, which is important for sUAS-based remote sensing since it can substantially reduce the level of effort required for collection setup and subsequent data analysis, and provide a means for large-scale automation of remote sensing data collection in all flight conditions.

## Figures and Tables

**Figure 1 sensors-23-00320-f001:**
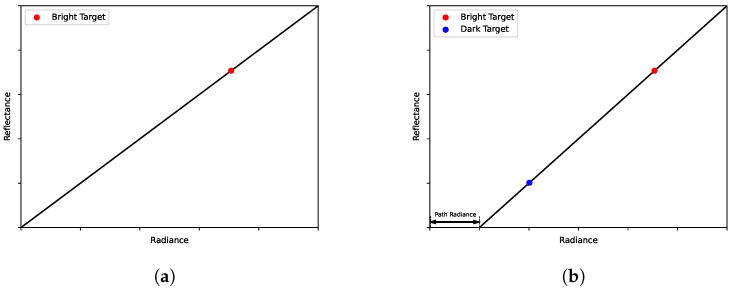
Arbitrary examples of the ELM relationship for a given spectral channel: (**a**) 1-point ELM. (**b**) 2-point ELM.

**Figure 2 sensors-23-00320-f002:**
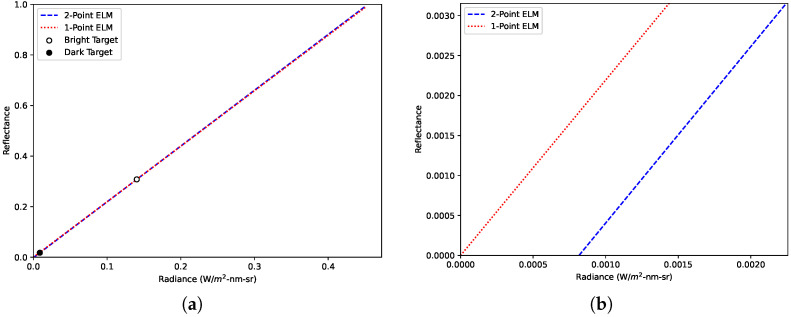
ELM relationships calculated for a data collection under clear conditions for a chosen wavelength of 555 nm: (**a**) ELM relationships from 0 to 1 reflectance units. (**b**) Zoomed plot from (**a**) highlighting path radiance.

**Figure 3 sensors-23-00320-f003:**
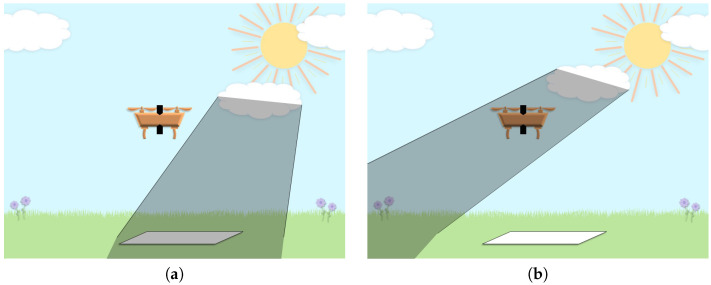
Diagrams illustrating situations where the irradiance on the ground would be different than at the sUAS: (**a**) Cloud casting a shadow on a target in the scene, but not on the sUAS. (**b**) Cloud casting a shadow on the sUAS, but not on the target in the scene.

**Figure 4 sensors-23-00320-f004:**
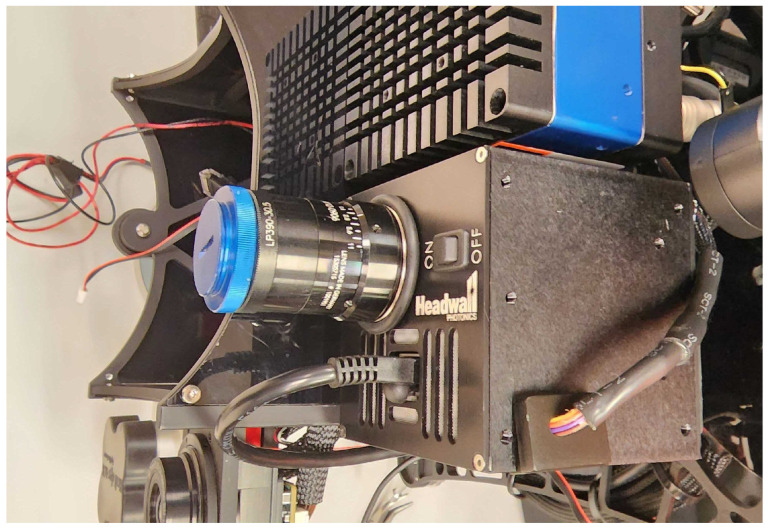
Headwall Nano-Hyperspec mounted on the MX2 sensor suite.

**Figure 5 sensors-23-00320-f005:**
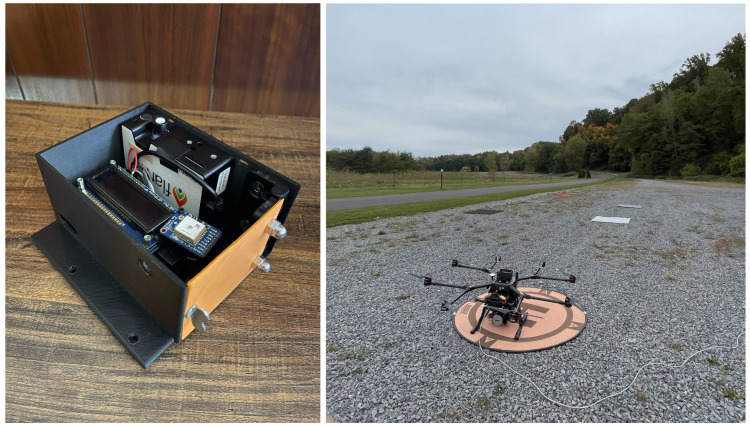
RIT DIS prototype, version 2, up-close and mounted on sUAS.

**Figure 6 sensors-23-00320-f006:**
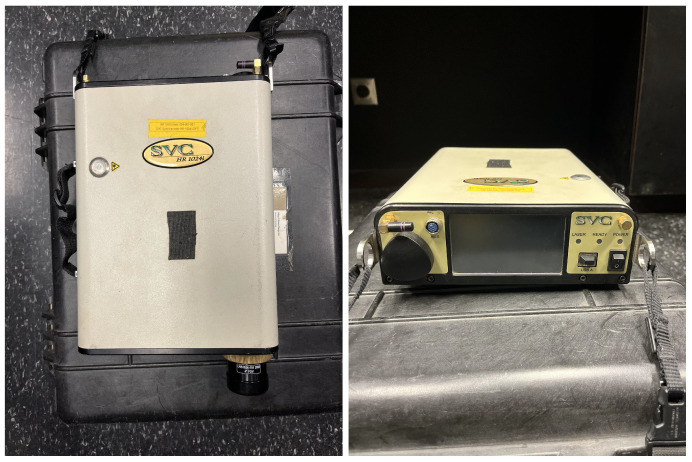
SVC HR-1024i field spectrometer.

**Figure 7 sensors-23-00320-f007:**
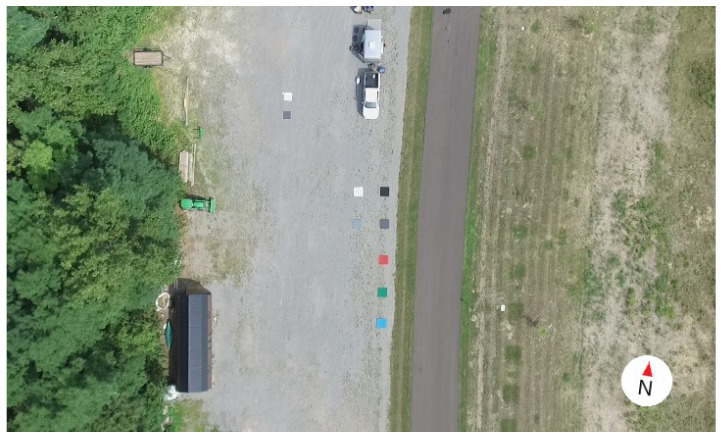
RGB image from sUAS of the target scene at Tait Preserve, depicting the organization of the scene for every data collect. The felt covered reference panels (red, green, blue, and 4 shades of gray) are located in the center of the scene.

**Figure 8 sensors-23-00320-f008:**
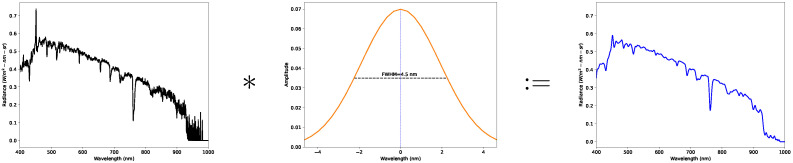
Effects of convolving the downwelling measurement with the blurring kernel.

**Figure 9 sensors-23-00320-f009:**
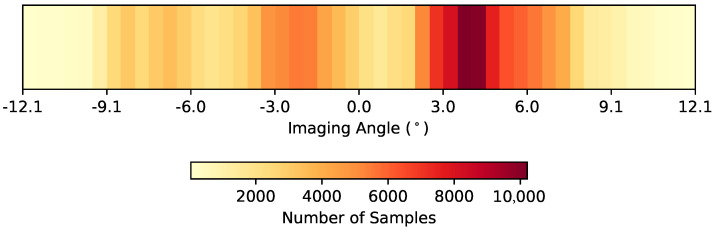
Number of target panel samples at various imaging angles across the three data collects in this study. The bins are 0.5° wide, and 0° represents nadir.

**Figure 10 sensors-23-00320-f010:**
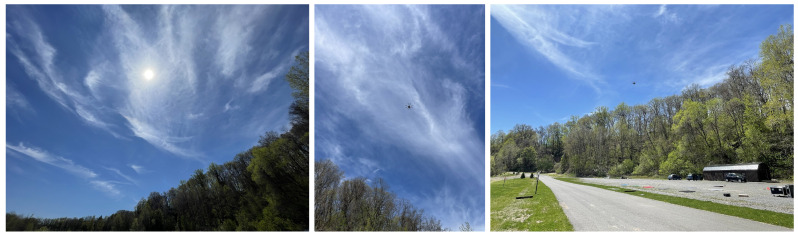
Photos taken at the data collection site on 11 May 2022 which illustrate the observed lighting and weather conditions.

**Figure 11 sensors-23-00320-f011:**
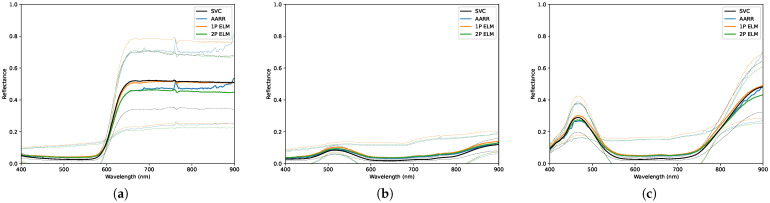
Distributions of reflectance measurements obtained, under light cloud conditions, using each conversion method for the: (**a**) Red panel. (**b**) Green panel. (**c**) Blue panel. The dashed lines designate a 2-standard deviation envelope.

**Figure 12 sensors-23-00320-f012:**
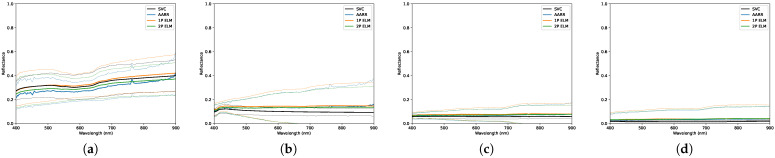
Distributions of reflectance measurements obtained, under light cloud conditions, using each conversion method for the: (**a**) Light gray panel. (**b**) Medium gray panel. (**c**) Dark gray panel. (**d**) Black panel. The dashed lines designate a 2-standard deviation envelope.

**Figure 13 sensors-23-00320-f013:**
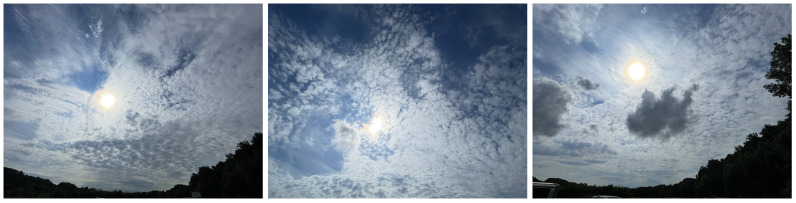
Photos taken at the data collection site on 16 August 2022 which illustrate the observed lighting and weather conditions.

**Figure 14 sensors-23-00320-f014:**
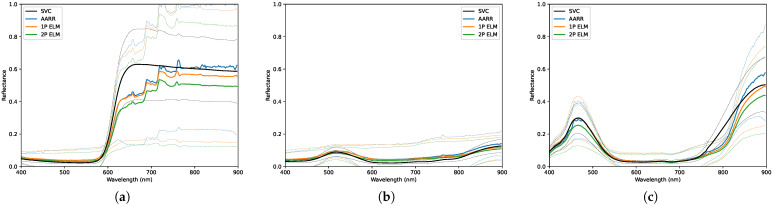
Distributions of reflectance measurements obtained, under scattered cloud conditions, using each conversion method for the: (**a**) Red panel. (**b**) Green panel. (**c**) Blue panel. The dashed lines designate a 2-standard deviation envelope.

**Figure 15 sensors-23-00320-f015:**
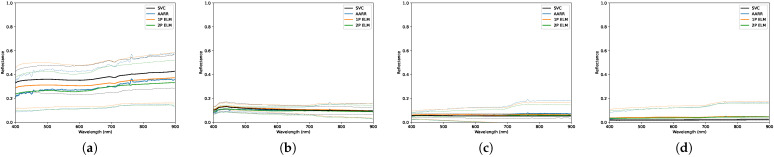
Distributions of reflectance measurements obtained, under scattered cloud conditions, using each conversion method for the: (**a**) Light gray panel. (**b**) Medium gray panel. (**c**) Dark gray panel. (**d**) Black panel. The dashed lines designate a 2-standard deviation envelope.

**Figure 16 sensors-23-00320-f016:**
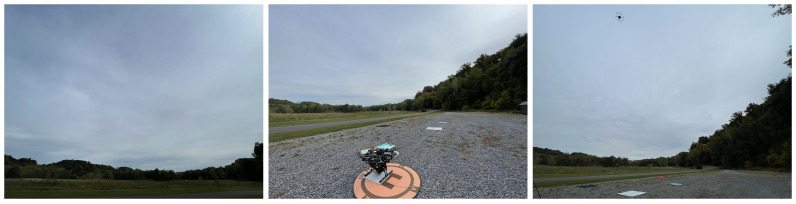
Photos taken at the data collection site on 12 October 2022 which illustrate the observed lighting and weather conditions.

**Figure 17 sensors-23-00320-f017:**
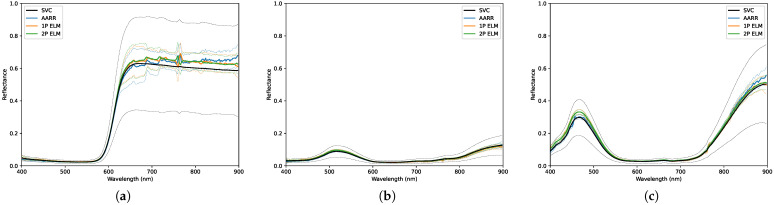
Distributions of reflectance measurements obtained, under overcast conditions, using each conversion method for the: (**a**) Red panel. (**b**) Green panel. (**c**) Blue panel. The dashed lines designate a 2-standard deviation envelope.

**Figure 18 sensors-23-00320-f018:**
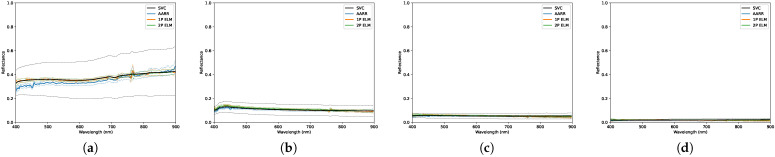
Distributions of reflectance measurements obtained, under scattered cloud conditions, using each conversion method for the: (**a**) Light gray panel. (**b**) Medium gray panel. (**c**) Dark gray panel. (**d**) Black panel. The dashed lines designate a 2-standard deviation envelope.

**Table 1 sensors-23-00320-t001:** Relevant specifications of the Headwall Nano-Hyperspec.

Wavelength Range	Spatial Bands	Spectral Bands	Pixel Dispersion
400–1000 nm	640	271	2.2 nm/pixel

**Table 2 sensors-23-00320-t002:** Relevant specifications of the Ocean Insight Flame-S-VIS-NIR-ES spectrometer integrated into the RIT DIS.

Wavelength Range	Spectral Bands	Pixel Dispersion
350–1000 nm	2027	0.38 nm/pixel

**Table 3 sensors-23-00320-t003:** Relevant specifications of the SVC HR-1024i field spectrometer.

Wavelength Range	Spectral Bands	Pixel Dispersion
350–2500 nm	1024	2.1 nm/pixel ^1^

^1^ Estimated average pixel dispersion; the HR-1024i utilizes 3 separate spectrometers of dissimilar spectral properties.

**Table 4 sensors-23-00320-t004:** Metadata for the first day of data collection, where light cloud conditions were observed.

Date	Time	Cloud Types
11 May 2022	16:30 UTC	Cirrostratus, Cirrus
**Solar Zenith**	**Temperature**	**Wind Speed, Direction**
26.3°	24.5 °C	1.34 m/s, NE

**Table 5 sensors-23-00320-t005:** $T^{2}$ statistic and ***RMSE*** for each conversion method’s reflectance estimates of each target panel reflectance compared to the ground-measured reflectance, as measured during the collect on 11 May 2022. $c_{0}$ is listed for each panel based on Equation (Equation 11).

Panel	Red	Green	Blue	Light Gray	Medium Gray	Dark Gray	Black
AARR	80.668	29.961	131.212	178.410	38.634	22.234	13.664
	** *±0.051* **	** *±0.018* **	** *±0.027* **	** *±0.045* **	** *±0.040* **	** *±0.017* **	** *±0.021* **
1-Point ELM	25.315	31.297	43.493	5.227	13.857	17.963	13.798
	** *±0.064* **	** *±0.022* **	* **±0.036** *	* **±0.056** *	* **±0.045** *	* **±0.020** *	* **±0.023** *
2-Point ELM	23.937	7.668	2.720	1.057	2.720	4.603	1.415
	* **±0.055** *	* **±0.018** *	* **±0.026** *	* **±0.036** *	* **±0.036** *	* **±0.016** *	* **±0.020** *
$c_{0}$	270.659	268.912	270.547	273.831	268.072	271.595	273.952

**Table 6 sensors-23-00320-t006:** $T^{2}$ statistic and ***RMSE*** for each conversion method’s reflectance estimates of each target panel reflectance compared to the ground-measured reflectance, as measured at an altitude of 30 m during the collect on 11 May 2022. $c_{0}$ is listed for each panel based on Equation (Equation 11).

Panel	Red	Green	Blue	Light Gray	Medium Gray	Dark Gray	Black
AARR	145.741	39.419	165.891	197.459	43.247	27.172	36.777
	** *±0.021* **	** *±0.013* **	** *±0.019* **	** *±0.021* **	** *±0.045* **	** *±0.014* **	** *±0.005* **
1-Point ELM	33.506	51.678	58.367	31.377	14.918	17.963	39.492
	** *±0.037* **	** *±0.016* **	** *±0.031* **	** *±0.047* **	** *±0.052* **	** *±0.015* **	** *±0.004* **
2-Point ELM	50.663	9.218	3.670	1.754	2.880	8.504	2.012
	** *±0.022* **	** *±0.013* **	** *±0.017* **	** *±0.012* **	** *±0.042* **	** *±0.012* **	** *±0.002* **
$c_{0}$	277.107	274.309	274.306	283.557	270.029	278.637	285.314

**Table 7 sensors-23-00320-t007:** $T^{2}$ statistic and ***RMSE*** for each conversion method’s reflectance estimates of each target panel reflectance compared to the ground-measured reflectance, as measured at an altitude of 90 m during the collect on 11 May 2022. $c_{0}$ is listed for each panel based on Equation (Equation 11).

Panel	Red	Green	Blue	Light Gray	Medium Gray	Dark Gray	Black
AARR	77.280	27.086	131.300	254.691	44.809	22.530	12.057
	** *±0.108* **	** *±0.026* **	** *±0.052* **	** *±0.082* **	** *±0.017* **	** *±0.023* **	** *±0.044* **
1-Point ELM	26.429	25.201	40.253	5.999	18.993	18.788	12.704
	** *±0.113* **	** *±0.032* **	** *±0.054* **	** *±0.069* **	** *±0.017* **	** *±0.029* **	** *±0.050* **
2-Point ELM	27.972	11.296	33.308	6.309	5.568	4.537	3.097
	** *±0.114* **	** *±0.027* **	** *±0.053* **	** *±0.072* **	** *±0.012* **	** *±0.023* **	** *±0.044* **
$c_{0}$	292.049	286.305	308.026	298.709	310.099	298.106	295.524

**Table 8 sensors-23-00320-t008:** Metadata for the second day of data collection, where scattered cloud conditions were observed.

Date	Time	Cloud Types
16 August 2022	15:00 UTC	Altocumulus, Cirrocumulus, Cirrostratus, Cirrus, Cumulus, Stratocumulus
**Solar Zenith**	**Temperature**	**Wind Speed, Direction**
41.2°	24.0 °C	2.24 m/s, N

**Table 9 sensors-23-00320-t009:** $T^{2}$ statistic and ***RMSE*** for each conversion method’s reflectance estimates of each target panel reflectance compared to the ground-measured reflectance, as measured during the collect on 16 August 2022. $c_{0}$ is listed for each panel based on Equation (Equation 11). Values in red indicate significant differences exist.

Panel	Red	Green	Blue	Light Gray	Medium Gray	Dark Gray	Black
AARR	254.787	30.383	2671.121	555.175	141.025	44.038	19.877
	* **±0.114** *	* **±0.023** *	* **±0.045** *	* **±0.085** *	* **±0.015** *	* **±0.021** *	* **±0.025** *
1-Point ELM	145.116	29.422	2869.807	5.249	17.890	14.423	12.469
	* **±0.116** *	* **±0.025** *	* **±0.049** *	* **±0.061** *	* **±0.013** *	* **±0.021** *	* **±0.028** *
2-Point ELM	463.502	14.065	3748.228	450.286	62.989	11.366	1.586
	* **±0.130** *	* **±0.024** *	* **±0.059** *	* **±0.095** *	* **±0.018** *	* **±0.021** *	* **±0.026** *
$c_{0}$	266.252	264.682	268.414	268.154	271.207	266.262	265.791

**Table 10 sensors-23-00320-t010:** $T^{2}$ statistic and ***RMSE*** for each conversion method’s reflectance estimates of each target panel reflectance compared to the ground-measured reflectance, as measured at an altitude of 30 m during the collect on 16 August 2022. $c_{0}$ is listed for each panel based on Equation (Equation 11). Values in red indicate significant differences exist.

Panel	Red	Green	Blue	Light Gray	Medium Gray	Dark Gray	Black
AARR	246.727	30.615	2028.649	488.046	127.480	44.010	18.823
	* **±0.115** *	* **±0.023** *	* **±0.044** *	* **±0.078** *	* **±0.016** *	* **±0.021** *	* **±0.022** *
1-Point ELM	121.050	29.974	3298.077	5.669	18.931	14.185	13.721
	* **±0.113** *	* **±0.025** *	* **±0.048** *	* **±0.053** *	* **±0.012** *	* **±0.020** *	* **±0.023** *
2-Point ELM	379.175	17.357	3457.804	434.683	72.347	12.135	1.163
	* **±0.129** *	* **±0.024** *	* **±0.059** *	* **±0.091** *	* **±0.018** *	* **±0.021** *	* **±0.021** *
$c_{0}$	266.727	265.401	269.336	269.225	272.589	266.576	266.457

**Table 11 sensors-23-00320-t011:** $T^{2}$ statistic and ***RMSE*** for each conversion method’s reflectance estimates of each target panel reflectance compared to the ground-measured reflectance, as measured at an altitude of 90 m during the collect on 16 August 2022. $c_{0}$ is listed for each panel based on Equation (Equation 11). Values in red indicate significant differences exist.

Panel	Red	Green	Blue	Light Gray	Medium Gray	Dark Gray	Black
AARR	403.257	42.401	2746.868	565.288	222.323	106.626	20.178
	* **±0.101** *	* **±0.018** *	* **±0.058** *	* **±0.145** *	* **±0.011** *	* **±0.022** *	* **±0.055** *
1-Point ELM	207.485	38.674	2568.569	8.912	33.385	41.249	17.305
	* **±0.166** *	* **±0.027** *	* **±0.070** *	* **±0.128** *	* **±0.024** *	* **±0.029** *	* **±0.073** *
2-Point ELM	170.661	16.517	2441.372	4.098	10.412	8.471	2.153
	* **±0.164** *	* **±0.025** *	* **±0.066** *	* **±0.128** *	* **±0.022** *	* **±0.027** *	* **±0.070** *
$c_{0}$	398.209	302.119	389.132	358.103	429.915	533.971	333.786

**Table 12 sensors-23-00320-t012:** Metadata for the third day of data collection, where overcast conditions were observed.

Date	Time	Cloud Types
12 October 2022	19:00 UTC	Altostratus, Cirrostratus, Stratus
**Solar Zenith**	**Temperature**	**Wind Speed, Direction**
58.0°	21.1 °C	5.81 m/s, S

**Table 13 sensors-23-00320-t013:** $T^{2}$ statistic and ***RMSE*** for each conversion method’s reflectance estimates of each target panel reflectance compared to the ground-measured reflectance, as measured during the collect on 12 October 2022. $c_{0}$ is listed for each panel based on Equation (Equation 11). Values in red indicate significant differences exist.

Panel	Red	Green	Blue	Light Gray	Medium Gray	Dark Gray	Black
AARR	113.231	33.967	253.465	271.712	112.790	34.326	10.768
	** *±0.032* **	** *±0.003* **	** *±0.014* **	** *±0.028* **	** *±0.005* **	** *±0.004* **	** *±0.003* **
1-Point ELM	45.753	148.448	349.451	236.441	69.753	71.467	90.717
	** *±0.028* **	** *±0.005* **	** *±0.016* **	** *±0.009* **	** *±0.007* **	** *±0.005* **	** *±0.003* **
2-Point ELM	25.951	74.243	204.540	229.78	34.771	21.388	2.668
	** *±0.030* **	** *±0.004* **	** *±0.015* **	** *±0.007* **	** *±0.006* **	** *±0.004* **	** *±0.002* **
$c_{0}$	270.581	268.140	266.399	276.028	290.033	274.175	265.797

**Table 14 sensors-23-00320-t014:** $T^{2}$ statistic and ***RMSE*** for each conversion method’s reflectance estimates of each target panel reflectance compared to the ground-measured reflectance, as measured at an altitude of 30 m during the collect on 12 October 2022. $c_{0}$ is listed for each panel based on Equation (Equation 11).

Panel	Red	Green	Blue	Light Gray	Medium Gray	Dark Gray	Black
AARR	63.655	27.970	101.233	217.648	71.213	31.866	9.034
	** *±0.023* **	** *±0.003* **	** *±0.010* **	** *±0.015* **	** *±0.004* **	** *±0.003* **	** *±0.003* **
1-Point ELM	46.443	29.716	65.261	240.240	86.570	18.930	11.682
	** *±0.029* **	** *±0.004* **	** *±0.010* **	** *±0.010* **	** *±0.009* **	** *±0.002* **	** *±0.002* **
2-Point ELM	23.591	16.881	35.973	221.946	25.529	4.111	2.182
	** *±0.032* **	** *±0.005* **	** *±0.006* **	** *±0.008* **	** *±0.006* **	** *±0.004* **	** *±0.003* **
$c_{0}$	271.619	270.012	266.753	279.797	296.328	278.138	265.990

**Table 15 sensors-23-00320-t015:** $T^{2}$ statistic and ***RMSE*** for each conversion method’s reflectance estimates of each target panel reflectance compared to the ground-measured reflectance, as measured at an altitude of 90 m during the collect on 12 October 2022. $c_{0}$ is listed for each panel based on Equation (Equation 11). Values in red indicate significant differences exist.

Panel	Red	Green	Blue	Light Gray	Medium Gray	Dark Gray	Black
AARR	142.655	54.766	706.389	569.483	397.389	58.167	69.198
	** *±0.033* **	** *±0.006* **	** *±0.020* **	** *±0.027* **	** *±0.008* **	** *±0.005* **	** *±0.004* **
1-Point ELM	172.697	149.577	736.312	562.413	170.115	52.004	62.606
	** *±0.025* **	** *±0.004* **	** *±0.018* **	** *±0.011* **	** *±0.012* **	** *±0.004* **	** *±0.001* **
2-Point ELM	142.996	140.918	239.155	602.443	61.406	14.556	165.314
	** *±0.028* **	** *±0.007* **	** *±0.017* **	** *±0.015* **	** *±0.005* **	** *±0.002* **	** *±0.001* **
$c_{0}$	479.649	313.247	496.083	385.026	761.488	347.089	866.265

## Data Availability

All relevant data can be requested by contacting corresponding authors.

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
