# Peer review of "Initial Performance Analysis of the At-Altitude Radiance Ratio Method for Reflectance Conversion of Hyperspectral Remote Sensing Data"

_sensors, 2022, doi:10.3390/s23010320_

Round 1

Reviewer 1 Report

In section 2.3, the discussion of radiance and irradiance really needs to be cleaned up to make sense - see the attachment for details. After that, I would like to review the modified manuscript. 

Having said that, I think that the results are valuable, especially given the increasing use of UAS and the need for reflectance derived from sensors on UAS. 

Author Response

Thank you for your review; please see attachment.

Reviewer 2 Report

In this interesting manuscript, a downwelling irradiance spectrometer integrated onto a small unmanned aircraft system is used to characterize the performance of AARR-generated reflectance from hyperspectral radiance data under a variety of challenging illumination conditions.

The possibility of converting reflectance in real time during data collection, the avoidance of inserting calibrated samples of pre-measured reflectance into the scene, and the consideration of variations in illumination conditions that AARR allows can facilitate large-scale automation of remote sensing data collection.

The article overall seems well written and worthy of publication. However, it would be desirable if the references were expanded to include more recent studies on the subject.

Asking for more detail in the description of the methods and presentation of the results, I suggest a re-reading to eliminate inaccuracies in the formatting, very few:

Figures 10, 11, 12, and 13, Line 401: Figures, as was correctly done for the others, should also be mentioned before being included in the main text. The same applies to Tables. 

Author Response

(The authors gave the same response as above.)

Round 2

Reviewer 1 Report

This version looks perfectly fine.